Divergent thermal specialisation of two South African entomopathogenic nematodes

Hill Matthew P. 1 hillmp@sun.ac.za
Malan Antoinette P. 2
Terblanche John S. 1
1 Centre of Excellence for Invasion Biology, Department of Conservation Ecology and Entomology, Faculty of AgriSciences, Stellenbosch University , South Africa
2 Department of Conservation Ecology and Entomology, Faculty of AgriSciences, Stellenbosch University , South Africa
Andrew Nigel
Electronic publication date: 2015 Jul 2
Publication date: 2015
Volume: 3
Electronic Location ID: e1023
Received 2015 Mar 30; Accepted 2015 May 26
Copyright: © 2015 Hill et al.
Copyright year: 2015
Copyright holder: Hill et al.
License: This is an open access article distributed under the terms of the Creative Commons Attribution License, which permits unrestricted use, distribution, reproduction and adaptation in any medium and for any purpose provided that it is properly attributed. For attribution, the original author(s), title, publication source (PeerJ) and either DOI or URL of the article must be cited.
License URL: https://creativecommons.org/licenses/by/4.0/

Keywords: Entomopathogenic nematodes, Thermal tolerance, Acclimation, Plasticity, Biocontrol

Funding: Technology and Human Resources for Industry Program THRIP TP2011060100026 Hortgro US-APE-NTT-2013-01 National Research Foundation Funding for this research was provided to JST through Hortgro grant US-APE-NTT-2013-01. JST and AM are also supported by the National Research Foundation THRIP award and Incentive Funding for Rated researchers. The funders had no role in study design, data collection and analysis, decision to publish, or preparation of the manuscript.

==============================
Thermal physiology of entomopathogenic nematodes (EPN) is a critical aspect of field performance and fitness. Thermal limits for survival and activity, and the ability of these limits to adjust (i.e., show phenotypic flexibility) depending on recent thermal history, are generally poorly established, especially for non-model nematode species. Here we report the acute thermal limits for survival, and the thermal acclimation-related plasticity thereof for two key endemic South African EPN species, Steinernema yirgalemense and Heterorhabditis zealandica. Results including LT50 indicate S. yirgalemense (LT50 = 40.8 ± 0.3 °C) has greater high temperature tolerance than H. zealandica (LT50 = 36.7 ± 0.2 °C), but S. yirgalemense (LT50 = −2.4 ± 0 °C) has poorer low temperature tolerance in comparison to H. zealandica (LT50 = −9.7 ± 0.3 °C), suggesting these two EPN species occupy divergent thermal niches to one another.

Acclimation had both negative and positive effects on temperature stress survival of both species, although the overall variation meant that many of these effects were non-significant. There was no indication of a consistent loss of plasticity with improved basal thermal tolerance for either species at upper lethal temperatures. At lower temperatures measured for H. zealandica, the 5 °C acclimation lowered survival until below −12.5 °C, where after it increased survival. Such results indicate that the thermal niche breadth of EPN species can differ significantly depending on recent thermal conditions, and should be characterized across a broad range of species to understand the evolution of thermal limits to performance and survival in this group.

Introduction

Temperature plays a key role in both the survival and activity of terrestrial invertebrates. Thermal tolerance may be characterized through traits such as thermal maxima and minima (absolute limits), processes or rates (e.g., development), as well as the optima thereof (e.g., temperatures for which growth rate and reproduction are maximised). Such basal thermal tolerance of terrestrial invertebrate species may also be adjusted through plastic responses induced through acclimation, hardening or acclimatization, allowing for phenotypic flexibility in the species relationship to temperature (Chown & Terblanche, 2006; Angilletta, 2009). Plasticity may thus allow for environmental extremes to be buffered against, although sometimes this comes with a trade-off with basal thermal tolerance (Calosi, Bilton & Spicer, 2008; Nyamukondiwa et al., 2011). Information on both basal thermal tolerance and the plasticity thereof can be compared to give insight into hierarchical levels of, and the magnitude and direction of, variation that may exist between species and across different groups (e.g., Hoffmann, Chown & Clusella-Trullas, 2013; Faulkner et al., 2014), and thus constraints or trade-offs that may be significant for understanding adaptive evolution.

Entomopathogenic nematodes (EPNs) are soil inhabiting insect parasites that fall within two monogeneric families: Heterorhabditidae and Steinernematidae (Rhabditida) (Hunt, 2007). The families do not share a common ancestry, but have nonetheless developed a similar lifestyle (Blaxter et al., 1998). Heterorhabditids and steinernematids each have corresponding mutualistic bacteria in the genera Photorhabdus and Xenorhabdus, respectively. These bacteria are held within the nematode intestine and released once the infective juvenile (IJ) has penetrated the insect host through the natural openings (spiracles, mouth or anus) (Griffin, Boemare & Lewis, 2005). The bacteria suppress the host’s immune system, typically killing the host within 24–48 h, subsequently providing nutrition for the nematodes and their offspring (Gaugler, Lewis & Stuart, 1997; Griffin, Boemare & Lewis, 2005). The EPNs are then able to complete multiple (e.g., up to three) generations within the insect cadaver (depending on the size of the host) before releasing a new cohort of IJs to start a new cycle (Gaugler, Lewis & Stuart, 1997). This life history strategy has allowed for the development of different biocontrol programmes using soil and aerial application of the IJs on agricultural pest insects (Lacey & Georgis, 2012).

There appears to be a wide range of responses to environmental stress exhibited across EPN species and strains, including variation in desiccation tolerance and hypoxia tolerance, and freeze tolerance (Morton & García-del-Pino, 2009; Salame et al., 2010; Shapiro-Ilan, Brown & Lewis, 2014). Some of the variation exhibited for cold tolerance is most likely attributed to different adaptive strategies including cryoprotective dehydration, anhydrobiosis and freeze avoidance or tolerance strategies (Wharton, 2011), although the mechanisms underlying cold tolerance responses are typically the primary focus of investigation (e.g., Ali & Wharton, 2014). While EPNs are able to withstand adverse environmental conditions inside their host cadavers, the IJ stage is required to seek out new hosts and thus may be particularly vulnerable to environmental variability in this life-stage (Brown & Gaugler, 1997). Much of the work looking at thermal performance of EPNs has focused on cold tolerance and how this relates to long-term cold storage solutions, or overwinter survival of free-living nematodes in polar environments (Perry & Wharton, 2011), although there have been assessments of heat tolerance in EPNs (e.g., Shapiro, Glazer & Segal, 1996; Jagdale & Gordon, 1998).

In addition to variation in basal thermal tolerance, acclimation-related adjustments in tolerance may be substantial in nematodes. For example, higher temperatures experienced during propagation increase upper thermal limits, while diminishing lower thermal ones (Grewal, Gaugler & Shupe, 1996; Shapiro, Glazer & Segal, 1996; Jagdale & Gordon, 1998). Further, short term exposure to lower temperatures prior to freezing has been shown to enhance freezing survival (Ali & Wharton, 2013). Previous EPN cold tolerance work has included findings that Heterorhabditis bacteriophora Poinar, 1976 and Steinernema feltiae (Filipjev, 1934), Wouts Mrác˘ek, Gerdin & Bedding, 1982 (Rhabditida:Heterorhabditidae) have demonstrated an increase in freezing tolerance after acclimation to temperatures below their propogation temperature (Brown & Gaugler, 1996; Ali & Wharton, 2013). Thus, testing thermal tolerances and associated plasticity or physiological adjustments in response to propagation or short-term temperature treatments to estimate thermal niche breadth across EPN species should help identify potential trade-offs between basal and acute responses. From an evolutionary perspective, understanding whether specialization to a particular environment has evolved at the expense of poorer performance in another environment is a significant avenue for forecasting and managing climate change responses over longer timescales (see discussion in Angilletta, 2009; Gilchrist, 1995).

Heterorhabditis zealandica and Steinernema yirgalemense Tesfamariam, Gozel, Gaugler and Adams, 2005 have demonstrated high virulence against a range of pest insect species in South Africa, including the mealybugs, Planococcus ficus (Signoret) and P. citri (Risso) (Hemiptera: Pseudococcidae) (Van Niekerk & Malan, 2012; Le Vieux & Malan, 2013) and tortricid moths (De Waal, Malan & Addison, 2011; Malan, Knoetze & Moore, 2011). Heterorhabditis zealandica was originally described in New Zealand, and though it is not a common species, it has been reported from the USA, Lithuania, Russia, Australia and South Africa (see Malan, Nguyen & Addison, 2006). Steinernema yirgalemense was first described from Ethiopia (Nguyen et al., 2004), and is highly prevalent there (Mekéte et al., 2005), and also found in Kenya (Mekéte et al., 2005) with only one isolate from South Africa (Malan, Knoetze & Moore, 2011). Importantly, these two species have demonstrated higher efficacy in host mortality than other EPN species and thus provide two potential candidates for ongoing pest management programmes (Van Niekerk & Malan, 2012). There are however, important differences between the species that require further investigation. Steinernema yirgalemense is two times more tolerant to low levels of free water than H. zealandica and has been demonstrated to detect and infect P. citri hosts quicker (Van Niekerk & Malan, 2012). Heterorhabditis zealandica and S. yirgalemense may therefore have contrasting thermal tolerance profiles.

In this paper we aim to characterise and compare the lethal upper and lower temperatures for H. zealandica and S. yirgalemense by using an accurately controlled thermal stage. We extend on estimating the thermal niche breadth and examine whether short term induced thermal acclimation is able to alter basal thermal resistance in these EPN species and discuss potential evolutionary trade-offs.

Materials & Methods

Codling moth cultures

Recently the virulence of H. zealandica and S. yirgalemense on codling moth Cydia pomonella L. (Lepidoptera: Tortricidae) was demonstrated to result in above 90% mortality and rapid EPN development time, thus providing a suitable in vivo host for these EPN species (De Waal, Malan & Addison, 2011). We obtained C. pomonella eggs and diet from Entomon Technologies (Pty) Ltd, Stellenbosch, Western Cape province, South Africa. The codling moth diet consists of agar, carrageenan, yeast, wheat germ, brown bread flower, ascorbic acid, benzoic acid, nipagin and formalin (Stenekamp, 2011). After hatching, C. pomonella larvae were reared on this diet under diapausing conditions in temperature controlled cabinets (10:14h L:D photoperiod, 25 °C, 60% RH).

Nematode cultures

We used IJ of H. zealandica and S. yirgalemense (Malan, Nguyen & Addison, 2006; Malan, Knoetze & Moore, 2011) from Stellenbosch University stocks (isolates SF41 and 157-C respectively). Both of these species were originally collected within South Africa (SF41 from Patensie, Eastern Cape and 157-C from Friedenheim, Mpumalanga (Malan, Nguyen & Addison, 2006; Malan, Knoetze & Moore, 2011). Fifth instar C. pomonella larvae were used as host for culturing IJ of the two EPNs species. Development period was held at 25 °C and lasted for around 10 days. Emerging IJ were collected over a 3 day period using modified White traps (White, 1927) and stored at 14 °C in distilled water, in horizontally placed, vented culture flasks, until use. We quantified the IJs to 100 individuals per 50 µl prior to assays (Navon & Ascher, 2000). The experiments were repeated with fresh batches of recycled nematodes, in an attempt to control for any cohort effects (see Supplemental Information 1).

Experimental setup

All temperature survival assays were performed using the stage design of Hill, Chown & Hoffmann (2013). This was originally used to measure critical thermal limits of mites, but was suited to temperature assays of EPNs due to being an open-well design. Briefly, this stage is an aluminium double-jacketed block (100 mm ∗ 100 mm ∗ 10 mm) with 19 wells (5 mm diameter). The wells are arranged in a circle and the middle well was used as a control, with a thermocouple (type K) secured in place. A perspex lid covers this stage so that each well is kept sealed during experimentation. Samples can be easily pipetted to and from each well and thoroughly rinsed between replicates to prevent contamination.

This stage was connected to a thermoregulator controller and waterbath (Huber cc410wl) filled with either ethanol for low temperature assays or 50:50 propylene glycol:water for high temperature assays. The temperature controlled fluid was pumped through channels in this block and allows for accurate temperature control, and verified independently.

Upper lethal temperatures

As the number of nematodes taken in each pipette draw was quite variable, we took 100 µl of each quantified sample to ensure a minimum number of individuals (n > 50) and then nematode samples were transferred into individual wells in the thermal stage. We used two replicates of each species for each different temperature assay.

The initial temperature was set and held for 5 min at 25 °C before being quickly ramped up at 0.5 °C min−1 to the set high temperature (set at: 35, 37, 39, 41, 43 °C). After being held at this temperature for 60 min, the temperature was ramped back down to 25 °C at 0.5 °C min−1. While this rate is different from temperature change experienced in the field (for the temperate regions of collection for these EPN species, e.g., Nyamukondiwa & Terblanche, 2010), it still allows for physiological differences between species to be contrasted, and has been employed for characterising nematode tolerances elsewhere (e.g., Ali & Wharton, 2013; Ali & Wharton, 2014), thus maximising possibility of comparison between studies. While this was still relatively quick, ramping assays are likely to be more ecologically relevant than static assays, especially in capturing elements of the daily thermal cycles and whether these may pose survival limits (Terblanche et al., 2011).

Temperature was recorded every second during the run using a PicoLog TC-08 USB datalogger and PicoLog software. We took the observed temperature to be the average temperature for the period between 3000 and 4000 s as this allowed for stabilization of the waterbath away from ramping temperature times.

Following assays each sample was drawn from the well and placed in a 500 µl Eppendorf tube, together with 0.025 g/100 mL Meldola’s blue dye to stain dead cells (Ogiga & Estey, 1974). Nematode survival was scored 24 h post assay with the aid of a dissection microscope (Leica MZ7s). By using a combination of dye penetration, mechanical stimulus (probing with dissection needle) and nematode shape it was possible to record mortality with a high level of accuracy.

Lower lethal temperature

We conducted these assays as we did for upper lethal temperatures, but with a few changes. Experiments were started at 5 °C, held at this temperature for 5 min and then ramped down at 0.5 °C min−1 to the different set temperatures (set at: −5, −6, −7, −8, −9, −10, −11, −12, −13, −14, −15 °C). While other studies have required seeded freezing of samples (e.g., Ali & Wharton, 2014; Shapiro-Ilan, Brown & Lewis, 2014), our stage design allows for freezing to occur passively, instead of supercooling taking place. Freezing was observed to occur in the wells through detection of an exothermic release on the temperature recordings. The temperature was then raised back up at the same rate to 5 °C before the nematodes taken to be held at 25 °C for 24 h prior to scoring survival. Observed temperature was calculated as the average between 1,000 and 2,000 s from recordings of the thermocouple in the middle well. Survival was scored as for the upper lethal temperature assays.

Acclimation

To investigate the effects of induced acclimation on EPNs we placed samples of both species into different constant temperature incubators (and cold rooms) and held for 24 h prior to experimentation. In previous studies examining thermal tolerance in EPN species, it was found that periods of time ranging from a few hours through to two days was sufficient to induce an acute effect of acclimation, hence we chose a period of 24 h (see discussion). We held samples at 5, 20 and 30 °C and included two replicates of each acclimation treatment with two control temperature samples (25 °C) for both species. For the acclimation treatments, we selected a number of temperatures appropriate to each species based on results from upper lethal temperature (ULT) and lower lethal temperature (LLT) experiments (H. zealandica: LLT: −5, −6, −7, −13, −14, −15 °C, ULT: 35, 37, 39 °C; S. yirgalemense: LLT: −5, −6 °C, ULT: 35, 37, 39, 41, 43 °C). We also added the control replicate data from these acclimation experiments back to our ULT and LLT datasets to increase their sample size.

Statistical analysis

All statistical analyses were performed in R (version 3.1.2; R Core Team, 2014). For the upper and lower lethal temperature experiments, without acclimation treatments, we modelled percentage survival as a function of observed temperature for both species separately using generalised linear models (glmfunction) with a binomial distribution and a logit link function. We tested for over-dispersion using the dispmod (version 1.1) package and then rescaled deviance to 1 when necessary. We then calculated LT10, LT50 and LT90 temperature profiles for each species using the dose.p function in the MASS package.

For the acclimation experiments, we again considered each species separately and performed generalised linear models using binomial distributions and logit link function. In this case we used set temperature rather than observed temperature so that we could examine acclimation between cohorts, and then added acclimation treatment as an additional effect and an interaction term between acclimation and set temperature. Temperature and acclimation treatments were used as categorical variables and then Wald’s χ2 test in the “arm” package was used to test the significance of these predictors.

Results

Heterorhabditis zealandica and S. yirgalemense displayed contrasting lethal temperature responses (Fig. 1). Overall, S. yirgalemense displayed greater survival at higher temperatures than H. zealandica (simplified Wald z = − 4.48, p < 0.001) (Table 1). For lower lethal temperatures, this pattern was reversed: H. zealandica had higher survival at low temperatures than S. yirgalemense (simplified Wald z = 7.99, p < 0.001) (Table 1). These differences are also reflected in the lethal temperature values predicted by the generalized linear models (Table 2). The predicted curves for survival of the two species show distinct and contrasting responses to upper and lower lethal temperatures (Fig. 1).

Figure 1 Thermal performance curves for two entomopathogenic nematodes.

(A) Lower lethal temperatures (LLTs) as a function of percentage survival (0–1). Curves glm logit model fits. Green is Heterorhabditis zealandica, orange is Steinernema yirgalemense. (B) Upper lethal temperatures (ULTs) as a function of percentage survival(/10). Curves represent glm logit model fits. Green is Heterorhabditis zealandica, orange is Steinernema yirgalemense. See Table 1 for model summaries.

Table 1 Generalised Linear Model summary for the effect of temperature on upper (ULT) and lower lethal temperature (LLT) limits of Heterorhabditis zealandica and Steinernema yirgalemense.

	Effect	df	Estimate	SE	z	P	
LLT H. zealandicaa	Intercept	1	4.94	0.58	8.58	<0.001	
Temperature	1	0.51	0.64	7.94	<0.001	
S. yirgalemense b	Intercept	1	4.35	0.75	5.81	<0.001	
Temperature	1	2.25	0.34	6.56	<0.001	
ULT H. zealandicac	Intercept	1	51.13	6.65	7.69	<0.001	
Temperature	1	−1.39	0.18	−7.7	<0.001	
S. yirgalemense d	Intercept	1	37.22	8.02	4.64	<0.001	
Temperature	1	−0.91	0.2	−4.63	<0.001	
Notes.

a residual deviance = 45.51, df = 60.

b residual deviance = 52.66, df = 48.

c residual deviance = 31.26, df = 35.

d residual deviance = 23.76, df = 38.

For both species, acclimation of thermal limits in both species resulted in both negative and positive responses to experimental temperatures as seen in both lower and higher survival limits at different temperatures (Fig. 2). There was considerable variation between acclimation treatments, and the overall effect of acclimation and the interaction with temperature was non-significant (Table 3). For H. zealandica, ULT performance was increased by the 30 °C acclimation treatment, especially at the 39 °C test temperature, although this was not significant. There was no increase in survival at 41 °C test temperatures from any of the acclimation treatments. For S. yirgalemense, the 30 °C acclimation increased survival at 41 °C and even slightly at 43 °C, although this again was not significant.

Figure 2 Acclimation and lethal thermal tolerances for two species of entomopathogenic nematodes, Heterorhabditis zealandica and Steinernema yirgalemense.

Data represents both batches of experiments pooled together. Error bars reflect ±1 standard error, Survival measured between 0 (complete mortality) and 1 (complete survival). (A) Upper lethal temperatures and acclimation for Heterorhabditis zealandica. (B) Upper lethal temperatures and acclimation for Steinernema yirgalemense (C) Lower lethal temperatures and acclimation for Heterorhabditis zealandica. (D) Lower lethal temperatures and acclimation for Steinernema yirgalemense.

Table 2 Generalised Linear Model/logit model predictions for 10, 50 and 90% survival of the population (i.e., LT90, LT50, LT10) of Heterorhabditis zealandica and Steinernema yirgalemense at upper (ULT) and lower (LLT) temperatures.

	H. zealandica	S. yirgalemense	
% Survival	LT	SE	LT	SE	
ULT (°C)	
10	38.2	0.3	43.2	0.7	
50	36.7	0.2	40.8	0.3	
90	35.1	0.2	38.4	0.6	
LLT (°C)	
10	−14.0	0.8	−3.3	0.1	
50	−9.7	0.3	−2.4	0.0	
90	−5.4	0.5	−1.6	0.1	

Table 3 Wald’s χ2 test with categorical acclimation and set temperature variables for both Lower Lethal Temperature (LLT) and Upper Lethal Temperature (ULT).

The generalized linear model used a binomial distribution for survival data with a logit link function and deviance was rescaled to 1. Test not possible to be performed on S. yirgalemense LLT data.

	χ 2	df	P	
ULT				
H. zealandica				
Temp	15.0	2	<0.001	
Acclimation	0.34	3	0.95	
Acclimation × Temp	4.5	6	0.61	
S. yirgalemense				
Temp	26.1	4	<0.001	
Acclimation	0.51	3	0.92	
Acclimation × Temp	4.8	11	0.94	
LLT				
H. zealandica				
Temp	42.9	6	<0.001	
Acclimation	6.8	3	0.08	
Acclimation × Temp	23.0	15	0.08	

For lower lethal temperatures, H. zealandica individuals acclimated at 5 °C have decreased performance until the experimental temperatures dropped below −12.5 °C, when this acclimation treatment gave increased survival down to −15 °C (significantly so from 20 and 30 °C acclimations; χ12=6.0, p = 0.014 and χ12=6.4, p = 0.011, respectively). Acclimations treatments of 5 and 30 °C were significantly different from one another for H. zealandica (χ12=6.2, p = 0.013) and this was driven by the differences at −7 and −13 °C (χ12=8.3, p = 0.004 and χ12=6.2, p = 0.011, respectively), displaying both decreased and increased survival, respectively. This pattern was also observed for the 5 and 20 °C acclimation treatments, at the −7 °C test temperature the 5 °C acclimation treatment displayed significantly decreased survival from the 20 and (χ12=5.0, p = 0.025) and then at the −13 °C test temperature, the 5 °C acclimation treatment again gave significantly increased survival over the 20 °C treatment (χ12=6.0, p = 0.014). While there was insufficient data to test for acclimation and LLT for S. yirgalemense, there appears to be a well-defined threshold of survival around −5 °C, with temperatures below this resulting in complete mortality and acclimation treatments unable to elicit a shift this threshold.

In the replication of acclimation experiments we observed some significant differences between experiments (Supplemental Information 1). Particularly for H. zealandica ULTs, the second cohort displayed significantly higher survival at 37 °C (χ12=25.6, p = < 0.001; Supplemental Information 1). Likewise, at 41 °C for S. yirgalemense, the first cohort was significantly different from the second (χ12=5.3, p ≤ 0.021) and the acclimation response was also more evident in the first experiments (Supplemental Information 1). For H. zealandica LLTs the cohorts were significantly different at −5 °C (χ12=9.4, p = 0.002), −7 °C (χ12=4.3, p = 0.39) and −13 °C (χ12=10.9, p = < 0.001). While these differences in response between the experimental cohorts were significant, the pattern of interaction between acclimation and temperature was largely preserved within both cohorts and on the combined data of these experiments (Supplemental Information 1).

Discussion

Our results here demonstrate contrasting thermal tolerance profiles for two EPN species and support findings that thermal tolerance varies widely across different EPN species (e.g., Morton & García-del-Pino, 2009; Shapiro-Ilan, Brown & Lewis, 2014). The two contrasting thermal tolerance curves were obtained on isolates that have been kept under the same laboratory conditions for many generations (Malan, Nguyen & Addison, 2006; Malan, Knoetze & Moore, 2011) and thus the differences observed here are likely to be due to fixed genetic differences rather than recent environmental influences (i.e., acclimatization). However, inadvertent directional selection during laboratory propagation may have influenced the estimates of thermal tolerance for these species (Grewal, Gaugler & Shupe, 1996). The two species are also from separate monogeneric families within the Rhabditida and have thus have had disparate evolutionary histories. It has been proposed that Heterorhabditis species are endemic to warmer climates, whilst Steinernema species more common in temperate regions (see Grewal, Selvan & Gaugler, 1994); however, our estimates of tolerances from the laboratory-propagated lines did not reflect this. Further studies on field collected EPNs are therefore required to examine if patterns of thermal resistance are consistently different between the two families in line with biogeographical hypotheses.

While the overall effects of acclimation were complex, both species exhibited some improved survival after being acclimated at 30 °C for high temperature assays, and consequently, do not support evidence of a direct trade-off between plastic and basal ULT. Other EPN studies have identified that acute acclimation provides improved thermal performance for some species (e.g., S. carpocase and S. feltiae; Jagdale & Grewal, 2003) but not others (e.g., S. ribrave; Jagdale & Grewal, 2003), indicating that trade-offs and mechanisms may exist in EPNs. While there is currently little information on trade offs between basal and induced thermal limits in EPN systems, previous studies of Drosophila species have identified trade-offs between basal low-temperature tolerance and acute low-temperature plasticity, but at high temperatures increased basal tolerance was accompanied by increased plasticity (Nyamukondiwa et al., 2011). However, thermal acclimation may provide other costs and benefits to performance, such as increased resistance to desiccation or reduced fecundity or longevity, which were not the focus of investigation here (Kleynhans et al., 2014; Terblanche, 2014). While our period of 24 h for acclimation may be considered quite short, short-term acute responses to temperature regimes have been documented in EPNs. For instance, a few hours at 35 °C has improved both H. bacteriophora survival and infectivity at 40 °C (Selvan et al., 1996), one to two days acclimation has improved thermal tolerance in S. carpocase and S. feltiae (Jagdale & Grewal, 2003), and an overnight freezing treatment prior to LLT characterization has demonstrated improved survival in S. feltiae (Ali & Wharton, 2013). As we were examining responses of previously uncharacterized species, we selected a time frame for acclimation that should not overtly stress the organism prior to treatment, but be sufficient to elicit a response. The onset and reversal of acclimation responses are not well understood even in more well-examined groups such as insects (Weldon, Terblanche & Chown, 2011), thus future tests should explore these responses further in EPNs across a broader range of conditions and time-scales.

Unlike most EPN species, which are freeze tolerant, H. zealandica has been shown to exhibit a freeze avoidance strategy, enabled through a protective sheath (Wharton & Surrey, 1994), and cold acclimation has been found to decrease freezing survival in H. zealandica (Surrey, 1996). We also found evidence for acclimation to decrease survival at some low temperatures on the isolate used: however, at the lowest temperatures the 5 °C acclimation treatment actually increased survival. In addition to the presence or absence of a protective sheath which greatly influences H. zealandica survival at low temperatures (Wharton & Surrey, 1994), species that undergo anhydrobiosis require that dehydration is actually possible through differences in water vapour pressure between the surrounding ice and bodily fluids (Holmstrup, Bayley & Ramløv, 2002). Anhydrobiosis has evolved as a freeze avoidance strategy in addition to supercooling in many soil invertebrates (Holmstrup, Bayley & Ramløv, 2002), and as the IJs were in an aqueous solution, such protective dehydration could probably not occur in the liquid medium we used for our assays, which may have influenced survival results for low temperatures tested here. Variation in results may also arise from differences in how acclimation and freezing are implemented into experiments, as substantial variation in results of acclimation experiments appear to be common for EPNs (Ali & Wharton, 2013). Differences from previous studies may also be related to our thermal stage design and reliance on freezing to occur spontaneously, as opposed to samples being seeded (inoculated) with ice crystals in other studies (e.g., Ali & Wharton, 2013). Ali & Wharton (2013) also suggest that the duration for which IJs are exposed to the temperature of ice nucleation has implications on survival of freeze-tolerant species such as S. feltiae. Experimental setup and freeze avoidance strategies are thus both likely to introduce further variation and have implications for species comparisons, especially for species where cryoprotective dehydration needs to occur (Brown & Gaugler, 1998; Ali & Wharton, 2013). This is something that should be considered in broader comparisons of cold tolerance across EPN species.

The differences between the cohorts in the replication of our acclimation experiments could be due to a number of factors. Given that nematodes are propagated under controlled conditions, the most likely sources of variation are perhaps differences in age of the nematodes on a given experimental day and/or variation in host health and nutritional content, e.g., lipid composition (Grunder & Jung, 2005) which may have a marked effect on plasticity and basal tolerance, especially if linked to IJ ability to synthesise heat shock proteins and cryoprotectants such as trehalose and glycerol (Grewal & Jagdale, 2002; Jagdale, Grewal & Salminen, 2005). Observer bias is also unlikely as a consistent directional influence since observers were well trained and produced inconsistent differences/similarities in direction of effects on the same treatment (see e.g., discussions in Terblanche et al., 2011; Castaneda et al., 2012; Blackburn et al., 2014). The cultures are maintained in a highly stable environment so environmental differences in terms of temperature and humidity are unlikely. Also, while we only conducted experiments on IJs that were less than 2 weeks old, the age of the IJ EPNs prior to experiment should be further investigated. We attempted to use all EPNs at the same time, but our experiment setup is limited to one stage and thus we are required to run our experiments in series, spanning over two weeks to complete them all. Time post emergence may be an important factor in thermal stress resistance.

Thermal tolerance of EPN species plays a key role in components of its life history, including mobility of the IJ, infection, development and reproduction (Grewal, Selvan & Gaugler, 1994; Morton & García-del-Pino, 2009; Salame et al., 2010; Shapiro-Ilan, Brown & Lewis, 2014). These results encourage for more EPN species to be characterized using these methods to better understand thermal tolerance in these nematodes and how this relates to biogeographical patterns as well as the evolution of thermal resistance in this group.

Supplemental Information

Supplemental Information 1 Acclimation and lethal thermal tolerances for two species of entomopathogenic nematodes, Heterorhabditis zealandica and Steinernema yirgalemense

Error bars reflect ±1 standard error. Data represents the cohorts seperately: red indicates the first batch of experiments scored by MPH, blue indicates the second cohort of experiments scored by EL.

Click here for additional data file.

The authors thank Elizabeth Louw for assisting with experiments. Md Habibullah Bahar and C Jaco Klok provided useful comments during review of this manuscript.

Additional Information and Declarations

Competing Interests

Author Contributions

Data Deposition

The authors declare there are no competing interests.

Matthew P. Hill conceived and designed the experiments, performed the experiments, analyzed the data, wrote the paper, prepared figures and/or tables, reviewed drafts of the paper.

Antoinette P. Malan conceived and designed the experiments, contributed reagents/materials/analysis tools, wrote the paper, reviewed drafts of the paper.

John S. Terblanche conceived and designed the experiments, analyzed the data, contributed reagents/materials/analysis tools, wrote the paper, reviewed drafts of the paper.

The following information was supplied regarding the deposition of related data:

Datasets for this study are available on FigShare:

http://dx.doi.org/10.6084/m9.figshare.1356153

http://dx.doi.org/10.6084/m9.figshare.1356152

http://dx.doi.org/10.6084/m9.figshare.1356151

http://dx.doi.org/10.6084/m9.figshare.1356150.

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
