# Peer review of "Divergent thermal specialisation of two South African entomopathogenic nematodes"

_PeerJ, doi:10.7717/peerj.1023_

## Round 0.1 · original submission · Minor Revisions

An interesting and well written and undertaken study. Both reviewers have made substantive comments in order to assist in the revision of this manuscript. Their concerns are primarily to enhance the readability of the manuscript, rather than substantive changes to it. Please address these comments in your revisions.

·

Basic reporting

Overall the manuscript is sound, however it needs major revision before publishing in PeerJ.

Experimental design

Good. but the list of temperatures used in this study is missing.

Validity of the findings

n

Additional comments

This study determines and ompare lethal upper and lower temperatures for two entomopathogenic nematodes and the impact of thermal acclimation on them. The overall experimentation, statistics and explanation are well enough to publish in PeerJ, however it needs some major revision before publishing.

Abstract: Abstract need to be more informative. I would suggest to include some numeric values of your results in abstract.
Line 13: name the two nematodes here.
Introduction:
Line 39-40: need a reference.
Materials and Methods: This section needs major attention to make it clearer to general audience.
- Details description of codling moth culture including details of diet are needed, or at least a good reference.
- There is no indication of the temperatures used in this experiment, the list of temperatures must be provided either in text or in a table
Line 101-103: unnecessary
Line 113: horizon
Line 127: …fluid was pumped [check the whole MS for grammar]
Line 130-131: not clear
Line 136: where is “this region”? explain it
Line 142-145, line 152-154, line 166: where are the temperatures used in this study?
Line 172: delete extra ()

Results: along with p value, it is better to add other statistical values i.e. F or chi square and df.
Discussion: needs additional explanation of this result with previous similar studies.
Line 235-237: this is example from insect study. There are tonnes of studies with insects. I think you should blend some studies on EPN and relate to this study.

·

Basic reporting

Hill and co-authors present an interesting exploratory study into the thermal tolerances of two nematode species of potential agricultural importance – as pest management agents, not pests themselves.
Their findings show two contrasting thermal profiles for these species but further investigations show that there may be a more complex set of strategies involved in these species’ thermal biology.

Experimental design

Following from the concise but comprehensive Introduction the Methods are clear in how the authors approached their study. They performed a series of standard and generally reliable experimental treatments to assess survival. Given the relatively unknown field of nematode thermal ecology this is basically the best way to make any original initial contributions to this field.

My only concern is the brief acclimation period – 24 hrs – they used for the acclimation studies. Nowhere do the authors justify this short period, or state results from preliminary work where they have determined that most acclimatory responses stabilize after the first 24 hrs. Some explanation of this would be useful.

Validity of the findings

The initial results were definitely intriguing. The contrasting thermal profiles of the two species as illustrated in Figure 1 would almost suggest that these species could be used across a broad seasonal period in the management of various insect pest species. However, their follow-up results, specifically from the acclimation studies, present several counter-intuitive outcomes. Specifically the LLT survival patterns – decreased survival at higher sub-zero temperatures followed by increased survival at lower sub-zero temperatures. Ecologically these findings just do not make sense. The authors are fortunately aware of this and have attempted to propose potential reasons for this in the discussion. However, already in the Introduction they have mentioned that in some nematode species anhydrobiosis might be a possible strategy for surviving cold periods. Since this requires significant levels of dehydration I find it surprising that the authors did not consider the possibility that their LLT procedure – cooling and freezing the animals in an aqueous medium – might not have been the most appropriate approach to follow in this regard. The possibility of anhydrobiosis would suggest that in preparation for cold periods the environments in which the animals may find themselves prior to cold exposure may not be waterlogged. This may be an additional source for the inconsistent results found.
This is not necessarily a knock against the study – given the novelty of the field one is bound to run into difficulties at first in selecting the most appropriate experimental procedures. It would, however, improve the study if the authors can address this more elaborately in the discussion.

This is, however, the first in hopefully a more comprehensive series of papers on these species and I am confident that this team of researchers will build and expand on what they have learned from these initial studies.

Additional comments

In general I quite liked this manuscript. The authors asked very basic questions and proceeded to address these in a standard and trusted series of experiments. But instead of easy run of the mill answers they have actually uncovered a slightly more complex system.
This paper should be ready of publication once the authors clarify and elaborate a bit more on the points mentioned above.

Also take note, I have made a small number of annotations in an attached copy of the reviewed manuscript (pdf). These should be addressed in concert with the comments made above.

Thank you for the opportunity to review this interesting study.

---

## Round 0.2 · accepted · Accept

You have addressed all the issues raised by the reviewers and I am happy for the manuscript to proceed to publication. Congratulations on the production of a well conceived and developed manuscript.